# OpenReview forum: "ProDA: Profile-Decomposed Adaptation for Capturing the Structured Residual in Low-Rank Updates"
_ICLR.cc/2026/Conference — ICLR 2026 Conference Withdrawn Submission_

### Official Review · Reviewer_2GeX · 2025-10-24

**Soundness:** 1
**Presentation:** 2
**Contribution:** 1
**Rating:** 2
**Confidence:** 4

**Summary:**

The paper proposes Profile-Decomposed Adaptation (ProDA) as an extension of LoRA. It introduces an additive “profile”
 $P = b_c \mathbf{1}^\top + \mathbf{1} b_r^\top$ and a “synergistic modulation” term, yielding the final update
$\Delta W = BA + P + \gamma(x) \odot (BA \odot P),$ where $\gamma(x)$ is a scalar produced by a small controller MLP and $\odot$ denotes elementwise multiplication.

The authors claim that this combination allows LoRA to capture “structured residuals” that low-rank updates miss. Empirically, the authors show that ProDA outperforms LoRA and other baselines.

**Strengths:**

- The proposed method is lightweight and easy to integrate into existing LoRA pipelines. The parameter overhead is minor and well-accounted for in the paper.

- The experiments cover both encoder and decoder architectures (RoBERTa, LLaMA-2/3) and span a diverse set of benchmarks (commonsense reasoning, GSM8K, GLUE, MT-Bench, HumanEval).

- The paper is generally clear and the motivation is easy to follow.

**Weaknesses:**

`W1: The core novelty largely collapses to a capacity increase (≈ higher-rank LoRA)`

The key “synergistic modulation” term expands algebraically to:
$$
 BA \odot P = (BA) \odot (b_c \mathbf{1}^\top) + (BA) \odot (\mathbf{1} b_r^\top)
 = \underbrace{\mathrm{diag}(b_c), BA}{\text{rank } r} + \underbrace{BA, \mathrm{diag}(b_r)}{\text{rank } r}.
 $$
Therefore,
$\Delta W = BA + P + \gamma \big( \mathrm{diag}(b_c), BA + BA, \mathrm{diag}(b_r) \big).$

Ignoring the weak additive term $P$, the modulation introduces two additional rank-$r$ LoRA-like components, tied by diagonal scales and a scalar gate. Effectively, this gives up to $3r$ rank capacity, just more constrained. So the “synergistic interaction” isn’t a new adaptation principle, it’s algebraically equivalent to a structured higher-rank LoRA. The paper doesn’t discuss this equivalence or compare against rank-matched LoRA baselines, which makes the improvements hard to attribute to the claimed “decomposition principle.” Most of the gains likely come from increased representational capacity, not from a fundamentally new mechanism.

`W2: Baselines are not rank- or capacity-matched`

If ProDA effectively triples LoRA’s rank (up to $3r$), then a fair comparison would include LoRA with $r' \approx 3r$$ or LoRA with simple per-feature scaling. Those baselines are missing. The paper evaluates LoRA and variants at rank 8, which is very low and a probable cause of their worse performance. Without these baselines, it’s impossible to know if the reported improvements stem from the new mechanism or just higher capacity.

`W3: Missing comparisons with high-rank LoRA variants`

The paper does not compare to LoRA variants that aim to capture higher-rank information such as HiRA (https://openreview.net/forum?id=TwJrTz9cRS), MoRA (https://arxiv.org/abs/2405.12130), and ABBA (https://arxiv.org/abs/2505.14238). These would be much stronger and more relevant baselines than the ones used in the paper.

`W4: The “structured residual” motivation is not supported by evidence`

The paper’s motivation, that LoRA misses a “structured residual” in fine-tuning updates, is never empirically shown. The singular value plots provided only demonstrate that LoRA is low-rank, which is expected by definition. There’s no analysis of full fine-tuning updates to confirm that the residuals indeed have an axis-aligned, low-dimensional structure that the profile could model. Without such evidence, the “structured residual” explanation feels speculative and unsubstantiated.

`W5: The additive profile is probably ineffective in pre-LN Transformers`

The proposed profile $P = b_c \mathbf{1}^\top + \mathbf{1} b_r^\top$ acts like adding a column-wise and a row-wise bias. But in modern pre-LN Transformers: The inputs are normalized, so $\mathbf{1}^\top x \approx 0$, making $b_c (\mathbf{1}^\top x)$ negligible.

The term $\mathbf{1} (b_r^\top x)$ adds the same scalar to all outputs, which the next LayerNorm mostly cancels.

These effects suggest that the additive profile has very limited expressive power. Given that, the large reported gains from this term alone (+6–10 points on some benchmarks) seem implausible. It’s more likely that differences in training setup or randomness explain those jumps, since the profile itself should be mostly neutralized by normalization.


---
These issues suggest that ProDA’s improvements likely stem from increased rank and capacity, not from the claimed new decomposition or structure. The paper’s motivation and theoretical story are not convincingly supported by either the math or the experiments.

**Questions:**

Please refer to weaknesses.

---

### Official Review · Reviewer_6yeQ · 2025-10-31

**Soundness:** 3
**Presentation:** 2
**Contribution:** 2
**Rating:** 2
**Confidence:** 3

**Summary:**

In my understanding, ProDA is an improvement over LoRA for fine-tuning large language models. LoRA assumes weight changes during fine-tuning are low-rank, but that’s the authors observed that this isn't true. There is a missing structured part in these changes, called the "profile". This is a structured "rank 2" matrix which is both added and modulates the low rank subspace. ProDA adds this profile term and even lets it interact dynamically with the low-rank update, and the Experiment section shows that the adaptation is now more expressive and accurate.

**Strengths:**

Novelty: The paper introduces a (somewhat) novel idea by improving over the low-rank assumption in PEFT and proposing the concept of a structured residual (“profile”) combined with dynamic modulation.

Experiments: The work is well-executed, with clear mathematical formulation and strong experimental validation across multiple benchmarks and architectures. It beats DoRA and PiSSA, which is very promising. The Ablation study helps show the contribution of each component.

Significance: ProDA achieves decent results while remaining highly parameter-efficient. In terms of significance, it is definitely impactful for practical large-scale model adaptation, as the LoRA overhead is not smuch.

**Weaknesses:**

W1. The paper does not compare ProDA against approaches stemming from Sparse High-Rank Adapters(SHiRA)[1] or S2FT[2] which also challenge the low-rank assumption in PEFT. These methods are widely explored across vision and language tasks, so a theoretical and experimental comparison would strengthen the paper. There is a lack of discussion on how ProDA differs conceptually from these masking-based approaches—both in terms of parameterization and adaptability.

W2. The writing is clear; There are no figures to help visualize the masks

W3. The paper lacks theoretical justification, and Subspace/Nullspace analysis.

1. S2FT: Efficient, Scalable and Generalizable LLM Fine-tuning by Structured Sparsity
2. Sparse High Rank Adapters
3. Zero-Shot Adaptation of Parameter-Efficient Fine-Tuning in Diffusion Models

**Questions:**

(For W1) Discuss how the masks are different from the structured masks already available; compare with PEFT methods which demonstrate structured/unstructured/sparse/high rank masks instead of LoRA-based appraoches.

(For W2) From a presentation perspective, the paper would benefit from a figure illustrating the shape and structure of these masks to help readers visually understand the contrast with ProDA’s profile-based design.

(For W3) Authors are encouraged to explore the "subspace analysis" section introduced in the LoRA paper. It talks in depth about information captured by LoRA which is orthogonal to the model. A thorough analysis of the amplification factor for LoRA vs. ProDA would help quantify what LoRA misses and what ProDA captures. ProLoRA[3] discusses the null space of LoRA weights. It would be valuable to investigate whether ProDA captures information lying in this null space. This will strengthen the theoretical foundation of this approach.

---

### Official Review · Reviewer_52gv · 2025-11-07

**Soundness:** 2
**Presentation:** 3
**Contribution:** 2
**Rating:** 4
**Confidence:** 4

**Summary:**

The paper presents a novel parameter-efficient fine-tuning method called Profile-Decomposed Adaptation (ProDA). The core idea is that the authors challenge the foundational assumption underlying Low-Rank Adaptation (LoRA) by arguing that weight updates during fine-tuning are not purely low-rank but contain a significant and highly structured residual component. The authors then propose ProDA, which models this residual using axis-aligned vectors (additive profile) and modulated interaction. These two terms are then integrated with the LoRA update to create a dynamic modulation mechanism. The experimental validation across commonsense reasoning, GLUE, and complex generative tasks demonstrates ProDA's benefits over LoRA, its variants, and even full fine-tuning in some cases.

**Strengths:**

$\bullet$  The paper identifies a genuine limitation in LoRA’s low-rank hypothesis that has largely gone unquestioned despite being foundational to PEFT research. The identification of the "delta profile" as a structured, exploitable signal rather than mere noise is insightful. Figure 1 effectively motivates this observation by showing the singular value spectrum, though the visual evidence could be strengthened with more rigorous statistical analysis across multiple layers and models.

$\bullet$ The experimental scope is impressive, covering diverse architectures (LLaMA-2, LLaMA-3 and RoBERTa models); multiple benchmarks (commonsense reasoning (8 datasets), GLUE benchmark, and generative reasoning (MT-Bench, GSM8K, HumanEval datasets)); and a number of baselines: DyLoRA, AdaLoRA, MELoRA, Delta-LoRA, DyLoRA, DoRA, PiSSA, rsLoRA, LoRA+ and  LoRA-GA.

$\bullet$ The two-principle design of ProDA is well-articulated. Moving from the additive profile (Principle 1) to the synergistic multiplicative modulation (Principle 2) represents a thoughtful progression. The use of a lightweight controller network to create input-dependent modulation sounds reasonable.

**Weaknesses:**

$\bullet$  **Limited Justification for the Delta Profile.**  While the motivation is intuitive, the paper lacks theoretical analysis of why the delta profile exists and what structural properties it possesses. Specifically, Section 3 lacks a theoretical or statistical justification (beyond Figure 1 and empirical discussion) for why this profile should be axis-aligned and efficiently parameterizable in all settings. This leaves open whether the observed efficacy is due to inherent properties of model weights or just an increased number of trainable parameters. For example, it would be greatly clarifying to provide visualizations or empirical characterizations of the learned axis-aligned profiles across several layers or tasks, to determine whether these profiles are consistent or simply a parameter-adding trick.

$\bullet$ **Limited Justification for the Modulation Mechanism.** In Section 3.3, the synergistic modulation relies on input-dependent gates, but the specific impact of the controller architecture (e.g. selection of bottleneck size $d_{bottle}$, number of layers, etc.) is not explored (two-layer MLP with bottleneck dimension is introduced without justification). The dependency of results on these hyperparameters, and on the gating function's form (e.g., $\gamma_\text{global}$ is not justified), is not ablated nor compared against possible alternatives.

$\bullet$ **Insufficient Analysis of Efficiency and Computational Overhead.** While the parameter count is discussed in Section 3.3, practical overheads (inference-time latency and memory overhead) are not quantified or compared to prior works at all. The statement that "the modulated interaction term introduces a minimal, input-dependent computational path" is not substantiated with any timing results. The statement “Given that $r$ and $d_{bottle}$ are small, $N_{ProDA}$ remains significantly smaller than the $d_{in} \times d_{out}$ parameters of the full matrix, preserving the core efficiency of PEFT methods” is irrelevant, as a fair comparison should be made across different PEFT methods. This raises critical concerns regarding the core claim of the paper (“We are the first to systematically demonstrate that the true weight update is a composite structure”), given that the experimental results may have been achieved solely due to an increase in the number of parameters.

$\bullet$ **Reproducibility Statement.** The authors promise code upon publication, but reproducibility is not currently verifiable, and there is no explicit availability of artifacts for reviewers (even just explicit hyperparameters values). For highly empirical contributions, this limits confidence prior to official release.

$\bullet$ **Potential Overstatement in Claims.** Previous weaknesses lead to potential overstatement in claims. Phrases such as "establishes a new state-of-the-art" and "suggesting it can act as a powerful regularizer" (Abstract, Conclusion, and Table 2/3 discussion) may be somewhat ambitious without the source code release and broader cross-domain evidence.

The paper's empirical results appear strong, but they don't validate the spectrum-based hypothesis—they only show ProDA works well as a PEFT method. The "delta profile" explanation remains speculative.

**Questions:**

1) Please address my concern regarding Figure 1. In lines 075-076, the authors claim “The spectrum of $\Delta W_{LoRA}$ confirms its explicitly low-rank nature by decaying sharply at the specified rank ($r = 128$)”, but the decay is not sharp at some specified rank, there are two distinct drops and they both happen at ranks $r < 64$.

2) Figure 1 truly demonstrates that LoRA tail is more structured than pure Gaussian noise and that there exists non-random organization in the neglected singular values, but this cannot prove that this structure is task-relevant, that this structure generalizes across models/tasks, that axis-aligned vectors can efficiently capture it or that the empirical gains come from exploiting this profile. Could you please clarify how Figure 1 explains this?

3) The claim that the profile is "highly structured" is stated but not rigorously characterized. What does it mean? It would be great to have some quantitative analysis besides Figure 1

4) Why does ProDA surpass full fine-tuning on GLUE? Is this reproducible across different random seeds and hyperparameter settings?

5) Have you considered different profile parameterizations?

6) What are the actual inference time and memory overheads in the experiments conducted?

7) Are there notable failure cases where ProDA provides limited or negative benefit compared to LoRA or full fine-tuning—for instance, on small-scale datasets, for tasks with low complexity, or when the low-rank assumption is in fact valid?

---

### Official Review · Reviewer_6vPi · 2025-11-07

**Soundness:** 2
**Presentation:** 3
**Contribution:** 1
**Rating:** 2
**Confidence:** 4

**Summary:**

The paper argues that the **fine-tuning weight update**
$
\Delta W = W_{\text{ft}} - W_0
$
is **not purely low-rank** (as assumed by LoRA), but contains a **structured residual** they call the *delta profile*.
They propose to **explicitly parameterize** this residual with a tiny number of additional parameters and then **combine it with the standard low-rank LoRA update**.
Empirically, they report consistent gains over LoRA and several variants across multiple benchmarks, and they provide ablations to attribute where the gains come from.

---

### Method

They start from the standard LoRA approximation:
$$
\Delta W \approx BA, \qquad
B \in \mathbb{R}^{d_{\text{out}} \times r} \;
A \in \mathbb{R}^{r \times d_{\text{in}}}.
$$

They then introduce a **delta profile** \(P\) to capture axis-aligned residual structure using **two vectors**:
$$
P = b_c \mathbf{1_{d_{\text{in}}}^{\top}} + \mathbf{1_{d_{\text{out}}}} b_{r}^{\top}
\qquad
b_c \in \mathbb{R}^{d_{\text{out}}}, \quad
b_r \in \mathbb{R}^{d_{\text{in}}}.
$$



This yields an **additive correction** to LoRA with only $\(d_{\text{out}} + d_{\text{in}}\)$ extra parameters.
They further propose a **synergistic (multiplicative) interaction** so that the profile can modulate the low-rank update in an input-dependent way:
$$
\Delta W_{\text{ProDA}}
= BA + P + \gamma(x) \odot (BA \odot P),
$$
where $\(\odot\)$ denotes the Hadamard (elementwise) product.



At inference, the static components can be merged into the base weights; the modulated term adds only a minimal input-dependent path.

---

### Experiments

**Setups.**
They evaluate on (i) **commonsense reasoning** (eight tasks), (ii) **GLUE** with RoBERTa-base, and (iii) **generative reasoning and coding** (MT-Bench, GSM8K, HumanEval) using LLaMA-2/3 and RoBERTa, training with standard Hugging Face / PEFT infrastructure.
Hyperparameters and training details (AdamW, warmup schedule, LoRA rank, dropout) are provided. The method shows good performance emperically, and the work also includes ablations to investigate the reason for improvement.

**Strengths:**

1) The paper presents a **well-executed experimental evaluation** across several benchmarks, including commonsense reasoning, GLUE, and generative reasoning tasks such as GSM8K and MT-Bench.   Across these tasks, the proposed method generally shows **consistent improvements over LoRA and a few of its variants** when trained with the same rank and comparable parameter budgets.


2) The approach itself is **simple and easy to understand**, extending LoRA with a small, interpretable modification.
The method is clearly described, and the ablation studies provide a reasonable analysis of how each component contributes to the performance gains.

**Weaknesses:**

While the **motivation** of revisiting LoRA’s low-rank constraint is valid, the **theoretical grounding** of the proposed solution is weak.
The paper argues that the residual beyond the low-rank subspace is structured based on smooth singular-value decay, yet this reasoning is unconvincing - **random matrices also exhibit smooth spectral tails**, so such behavior does not necessarily indicate recoverable structure. Thus, the claim that LoRA systematically misses structured high-rank components lacks rigorous justification.

A more substantial concern lies in the **effective rank** of the proposed delta profile. The modification introduces only a **rank-2 correction** to LoRA, while keeping the same parameter budget. This makes the method essentially equivalent to **LoRA with rank \(r+2\)**.
Although the Hadamard modulation theoretically adds rank, there is **no clear reason for this specific parameterization** to be more effective than simpler high-rank extensions.
The **ablation results exacerbate this issue**: the additive profile (\(BA + P\)) alone yields large gains (∼10 % on MATH), while the Hadamard modulation - supposedly the high-rank component - adds only marginal improvements. This contradicts the paper’s own motivation, since the main gains appear to stem from adding two low-rank directions rather than capturing structured high-rank signals.

Finally, the paper **omits direct comparisons to several highly relevant PEFT baselines** that already address LoRA’s rank limitation.
Notably, **HiRA** [1] introduces Hadamard-based modulation to achieve high-rank updates while retaining LoRA’s efficiency,
**MoRA** [2] enables high-rank updates through input compression and activation decompression,
and **ABBA** [3] models the update as a Hadamard product of two independent low-rank matrices, achieving full learnability and higher expressivity under the same parameter budget.
These works directly tackle the same expressivity bottleneck and should be discussed and compared against.
Their omission leaves a **major empirical and conceptual gap** in the evaluation.


## References

**HiRA: Parameter-Efficient Fine-Tuning with High-Rank Adaptation via Hadamard Product.**
Zixuan Ke, et al. *International Conference on Learning Representations (ICLR), 2025.*

**MoRA: Low-Rank Adaptation via Input Compression and Output Decompression.**
Qiang Zhang, et al. *ArXiv preprint arXiv:2402.10177, 2024.*


**ABBA-Adapters: Efficient and Expressive Fine-Tuning of Foundation Models.**
Raghav Singhal, Kaustubh Ponkshe, Rohit Vartak, and Praneeth Vepakomma.
*ArXiv preprint arXiv:2505.14238v3, 2025.*

**Questions:**

1. **Spectral analysis:**
   The motivation relies heavily on the idea that the residual update beyond LoRA’s low-rank subspace is structured.
   Could you provide the **singular-value spectrum of ProDA’s full update** $(\(BA + P + \gamma \odot (BA \odot P)\))$ compared to LoRA and full fine-tuning?
   This would help verify whether the proposed parameterization actually increases effective rank or captures distinct structure.

2. **Rank equivalence:**
   Given that the additive profile \(P\) introduces only a rank-2 correction, have you compared directly against a **LoRA model with rank \(r+2\)** under the same parameter budget?
   Such an experiment would help isolate whether the improvements come from true structural modeling or simply from extra rank.

3. **Choice of Hadamard modulation:**
   What motivated the specific **Hadamard (elementwise) interaction** for modulation rather than, for example, concatenation, gating, or bilinear fusion?
   Is there theoretical or empirical evidence suggesting that this particular form captures meaningful dependencies?

4. **Relation to high-rank PEFT methods:**
   Several recent methods - **HiRA**, **MoRA**, and **ABBA** - explicitly address LoRA’s limited expressivity and demonstrate near full-rank adaptation under similar budgets.
   Could you clarify how ProDA differs from or complements these approaches, and why these baselines were omitted from experimental comparison?

5. **Interpretation of ablation results:**
   The ablation suggests that \(BA + P\) accounts for almost all the performance improvement, while the modulation term contributes little.
   How do you reconcile this with your claim that the key benefit comes from modeling structured high-rank residuals?

---

### Note · Authors · 2026-01-02

I have read and agree with the venue's withdrawal policy on behalf of myself and my co-authors.